## Research Article

mental health; stigmatization; interventions

**Corresponding author:**
Manvi Poddar;
Email: mpoddar1@alumni.jh.edu

# Designing integrated care models for mental health and tuberculosis in Pune, India: A formative qualitative study of patient, caregiver and provider perspectives

Manvi Poddar[1] ⓘ, Madhuri Thorat Nalavade[2], Nishi Suryavanshi[2], Jonathan E. Golub[3], Judith Bass[4] ⓘ, Christopher G. Kemp[1] ⓘ and TB Aftermath Study Team[5,6]

[1]Department of International Health, Johns Hopkins Bloomberg School of Public Health, USA; [2]Johns Hopkins Center for Infectious Diseases, India; [3]Department of Medicine, The Johns Hopkins University School of Medicine, USA; [4]Johns Hopkins University Bloomberg School of Public Health, USA; [5]The Johns Hopkins University School of Medicine, USA and [6]Dr D Y Patil Medical College Hospital and Research Centre, India

## Abstract

People with tuberculosis (TB) and TB survivors are at increased risk for mental health (MH) conditions. Better management of conditions like depression can improve adherence to TB treatment, and integrating MH care into TB treatment may reduce the MH treatment gap and improve outcomes. This qualitative study explored design characteristics for integrated MH-TB care in Pune, India. Data collection involved in-depth interviews (n = 25) with TB survivors with lived experience of MH conditions, their family members, and TB and MH providers. Data collection and analysis were guided by the Consolidated Framework for Implementation Research, and journey maps illustrated patient experiences. Participants shared suggestions for integrated care models, advantages and barriers to integration, intervention delivery agents, and local perceptions of MH conditions. Barriers included limited awareness about MH and perspectives about MH treatment, which were limited to consuming medication. Suggestions for integrated interventions included raising awareness about MH conditions and existing MH services among TB providers, regular MH screening and counseling for people with TB, and engaging TB survivors to share their experiences with patients in group settings. These insights highlight the importance of working with people with lived experience and understanding patient journeys to inform intervention implementation and sustainability.

## Impact statement

This research elucidates the perspectives of patients, caregivers and healthcare providers regarding integrated models for mental health (MH) and tuberculosis (TB) care in India. The findings possess significant potential to inform program and policy development within the Indian healthcare landscape. By incorporating a variety of lived experiences, the study offers valuable insights into the barriers and facilitators associated with seeking MH and TB care, while also underscoring challenges faced by health providers in providing care. The recommendations derived from this research can enhance the design and delivery of interventions within India's health system, while effectively integrating the perspectives of patients.

## Introduction

Globally, in 2023, 1.25 million people died from tuberculosis (TB) (World Health Organization, 2024), with India accounting for 25% of the global 10 million TB cases (Central TB Division, Ministry of Health and Family Welfare, 2023). TB patients experience treatment gaps along the full TB care cascade, ranging from limited access to diagnostic facilities, misdiagnosis, delayed treatment initiation and lost to treatment follow-up (Subbaraman et al., 2020). In addition to challenges related to treatment adherence and access, people with TB (PWTB) and TB survivors are at increased risk for common mental disorders, such as anxiety or depression (Agbeko et al., 2022). A recent scoping review found that TB and its associated comorbidities increase the risk for depression by three to six times (Agbeko et al., 2022). The prevalence of mental health (MH) conditions, such as depression and anxiety, is elevated among PWTB (Janse Van Rensburg et al., 2020), and there is a strong association between depressive symptoms and negative TB outcomes, lost to TB treatment follow-up and death (Ruiz-Grosso et al., 2020).

Globally, one in eight people were living with mental disorders in 2019 (World Health Organization, 2025). Anxiety and depression were among the most common. MH conditions, such as depression among PWTB, can lead to the delayed initiation of seeking TB treatment (Hayward et al., 2022). A study from India showed that one in three people on drug-resistant TB treatment experienced psychiatric comorbidity (Laxmeshwar et al., 2022). Depression is found to particularly affect people with a multidrug-resistant TB diagnosis and people who are taking anti-TB medications (Dan-ni et al., 2024). Stigma related to certain chronic conditions, such as TB, is associated with increased psychiatric morbidity, namely anxiety and depression (Kane et al., 2019).

Better management of conditions such as depression could improve adherence to TB treatment (Pachi et al., 2013; Sweetland et al., 2017). A study (Janmeja et al., 2005) from India used psychosocial support groups to assess the role of behavior modification in improving compliance with short-course anti-TB chemotherapy. The study showed that psychological interventions are effective in improving adherence to treatment and reducing treatment failure, relapse and drug resistance (Janmeja et al., 2005). Integration of MH care into TB treatment has the potential to reduce the MH treatment gap and improve outcomes for both conditions (Collins et al., 2011; Pasha et al., 2021).

According to the National Survey of Mental Health Resources conducted in 2002, 3,800 psychiatrists existed in the country against the 11,500 needed (Garg et al., 2019). The 2015–2016 National Mental Health Survey, conducted across select states in India, found that the number of psychiatric social workers was notably low across all surveyed states. In the state of Kerala, which had the highest number among the surveyed states, there were only 0.6 clinical psychologists per 100,000 population (National Institute of Mental Health and Neuro Sciences Bengaluru, 2016).

A study from Nepal engaged MH service users and health workers as 'cofacilitators' in a contact-based intervention to reduce stigma and improve MH services (Kohrt et al., 2018). Another study from India designed a digital training program for community health workers aimed at delivering psychosocial treatment for depression (Muke et al., 2020; Black et al., 2023). The intervention developed was tested for feasibility and acceptability, and it showed that digital training, coupled with added support, can be as effective as in-person training. Working with people who have lived experience can also promote the acceptability and sustainability of the integrated care package (Kohrt et al., 2021). Understanding the needs and experiences of people with lived experience of both TB and MH, along with their family members and caregivers, and both TB and MH providers, will support efforts to design an integrated intervention, and facilitate greater accountability and transparency in the opportunities and constraints of the local context (Singh et al., 2023).

## Study objective

The objective of this formative study is to understand the perspectives and needs of different relevant groups of people to identify key design characteristics of potential integrated care models for MH and TB in India. Relevant groups include individuals who have lived experience of MH issues and TB, along with their family members, MH providers and TB care providers.

## Methods

### Study setting

This exploratory qualitative study was embedded in TB Aftermath (Cox et al., 2022), a noninferiority randomized trial that is testing two active case finding strategies among TB survivors and their household contacts, and was supported by pilot funding specifically intended to explore lived experiences of MH conditions following TB treatment. TB Aftermath was based in Pune, a city in Maharashtra, India. The State of Maharashtra has a TB case notification rate of 183 cases per 100,000 people (Central TB Division, Ministry of Health and Family Welfare, 2023). The District TB Centre is located in Pune city, serving a population of 3.6 million people in rural areas and 2.7 million in urban areas (Atre et al., 2004). Each Tuberculosis Unit (TU) in Pune serves ~300,000–500,000 people (Cox et al., 2022).

### Theoretical framework

The Consolidated Framework for Implementation Research (CFIR) was used to guide data collection and analysis. CFIR comprises five domains: innovation, outer setting, inner setting, individuals and implementation process. It aims to predict barriers and facilitators of implementation effectiveness, and helps tailor implementation strategies (Damschroder et al., 2022).

### Data collection

Data collection occurred between June 2023 and December 2023. Data collection instruments were developed using the five CFIR domains. Purposive sampling was used to recruit participants in the study. A total of 25 in-depth interviews were conducted with TB survivors with concurrent MH conditions ($n = 10$), their household members ($n = 5$), TB providers ($n = 5$) and MH providers ($n = 5$). A sample size of $n = 25$ is consistent with established guidance, suggesting that 20–30 interviews are often sufficient to reach saturation and ensure depth and diversity of perspectives in qualitative research (Guest et al., 2006; Namey et al., 2016).

TB survivors and their household members were recruited from a pool of participants who were enrolled in the TB Aftermath Study. TB survivors who had completed their 18-month follow-up visit in the TB Aftermath Study were eligible for recruitment. From this pool of eligible participants, TB survivors with MH conditions were identified and recruited if they met one or more of the following criteria:

- MH condition identified through Patient Health Questionnaire-9 (PHQ-9) (Kroenke et al., 2001) (Depression) or Generalized Anxiety Disorder-7 (GAD-7) (Spitzer et al., 2006) (Anxiety), with a score of 10 or more,
- Self-reported MH condition at either the time of TB diagnosis or treatment initiation, enrollment into TB Aftermath or at the 18-month follow-up visit for TB Aftermath.

Participants who screened positive for common mental disorders were referred to the government hospital's psychiatry outpatient department as per the study protocol; however, due to the cross-sectional nature of this study, information on whether they accessed any MH services is not available.

Written informed consent was obtained before each in-depth interview. TB survivors and their family members were compensated for their time. All interviews were conducted in Hindi,

Marathi, English or a combination of these languages; participants expressed their language preferences before the interview. The primary interviewer (MTN) was accompanied by a field notetaker (SS and MP). TB survivors and their family members were interviewed in TUs or their homes. TB providers and MH providers were interviewed in TUs or at their place of work. The interviews spanned from 30 to 70 min.

All data collection activities were completed by MTN, MP and NS, who were trained in qualitative data collection and certified in human subjects research. Data collection concluded when interviews no longer yielded new emerging themes, indicating that data saturation had been reached within the sample of 25 participants (Sandelowski, 1995; Marshall, 1996; Hennink and Kaiser, 2022).

### Data analysis

Audio recordings of each interview were transcribed in the language in which the interview was conducted. The transcript was then translated from Marathi or Hindi into English. Participant identities were de-identified during transcription. All transcripts were read closely and coded on NVivo (version 1.7.1) using a combination of inductive and deductive codes (Supplementary Annexure 1).

This study used template analysis, combining deductive coding based on a framework with inductive coding to allow new themes to emerge from the data. Constructs and domains from CFIR were used to form deductive codes. The updated CFIR (Damschroder et al., 2022) was used as a foundation for deductive coding. Relevant constructs from the original CFIR (Damschroder et al., 2009) and recommendations for adapting CFIR for use in low- and middle-income countries (Means et al., 2020) were also integrated to tailor the framework to the context of integrated TB–MH care in India. Table 1 maps the selected CFIR constructs, corresponding domains and the themes identified during analysis. Inductive codes, such as 'patient journey and pathway' or 'MH perception', were developed where the CFIR codes did not apply.

Preliminary coding was completed across all transcripts by MP. CK, NS and MTN then selected a subset of transcripts, at random, to code independently and check for inter-coder reliability. Analytical memos were used to record emerging themes as each transcript was read and coded.

The analyzed data were used to develop patient journey maps that depict and visualize the MH and TB treatment experiences of TB survivors. A journey map is a visualization tool that offers insight into patient pathways to seeking care, experiences with interventions or the barriers patients face as they interact with a health system (Design for Health, n.d.). These maps were developed through researcher-driven analysis of recorded participant experiences. Portions of subcodes from the coded data were used to extract information related to patient experiences using an Excel sheet. Themes such as TB diagnosis and TB treatment were used to map out patient experiences. These extracted portions were consolidated and cross-tabulated to find similarities across different patient experiences. Two patient journey maps were developed, representing separate experiences for a male TB survivor and a female TB survivor with lived experiences of MH conditions.

Patient journey maps may help implementors think about implementation strategies along the cascade of TB care. The patient journey maps in this study highlight participant perspectives with TB and MH conditions as an early exploratory tool. The patient journey maps and findings from this study were presented back to the participants and will help inform a subsequent Phase 2 of this work that aims to develop a menu of strategies for MH-TB integration.

## Results

Table 2 summarizes the total number of participants enrolled in the study, with details about their median age and gender representation. Table 3 outlines characteristics of TB survivors and their family members. Of the total participants enrolled, 48% were females ($n$ = 12). TB survivors were drawn from three TB Care Units (TUs) in Pune: Sahkarnagar (50%), Jijamata (40%) and Paud (10%).

Participants spoke most about their ideas for integrated interventions, their experiences with TB and MH, and the advantages of implementing an integrated intervention, along with the complexities that must be taken into consideration. From these codes, the following key themes were identified:

- Integrated interventions (suggestions, advantages and barriers)
- Agent for intervention delivery
- Local attitudes and perceptions of MH conditions

### Integrated interventions

Participants shared their suggestions for interventions or programs that integrated MH and TB care in India, advantages that these integrated services may have for PWTB and society and the potential barriers to implementation that the interventions may face. Participant suggestions on the integrated interventions were coded inductively. Their perspectives on the potential advantages of the intervention and barriers were coded deductively using three CFIR constructs: Relative Advantage, Complexity and Knowledge and Beliefs.

#### Raising awareness

A key suggestion from participants included creating awareness about TB disease, MH conditions and services available to treat and manage these conditions. Participants believed that by raising awareness among PWTB, they may feel less tense about their TB diagnosis and will know how and where to seek care should they need MH services.

> 'Services for mental stress, means we have to advertise it. Like if you have mental stress, then you can come here and talk to us, you can share it with us, we need to assure them...' – TBS01

Participants also highlighted the importance of raising awareness about MH in larger community settings.

> 'Public awareness is most important. As we have discussed now that we are in school, this should be done in college also, in your gram sabha [community meeting], in that gram sabha, whenever the topic is related to health, we should keep a part of mental health there'. – TBP02

#### MH counseling

They also emphasized the need for and importance of counseling. MH providers explained that early counseling sessions (at the time of TB diagnosis) may help providers identify early warning signs of MH conditions.

**Table 1.** Deductive and inductive constructs used for coding and their corresponding themes

| CFIR domain | CFIR construct | Theme |
|---|---|---|
| Individual | Innovation deliverers[a] | Agent for intervention delivery |
| | Knowledge and beliefs[a] | Potential advantages of the intervention or barriers |
| | Self-efficacy[a] | Agent for intervention delivery |
| Inner setting | Available resources[a] | Institutional factors influencing program design and implementation |
| | Compatibility[a] | |
| | Culture[a] | |
| | Implementation climate[b] | |
| | Network and communication[b] | |
| | Organizational incentives and rewards[b] | |
| | Readiness for implementation[b] | |
| Outer setting | Community characteristics[c] | Factors influencing program design and implementation |
| | External policy and incentives[b] | |
| | Local attitudes[a] | Local attitudes and perceptions of mental health conditions |
| | Patient needs and resources[c] | Institutional factors influencing program design and implementation |
| Innovation | Adaptability[a] | Potential advantages of the intervention or barriers |
| | Complexity[a] | |
| | Cost[a] | |
| | Relative advantage[a] | |
| | Sustainability[c] | |
| Process | Decision-making[c] | Agent for intervention delivery |
| | Engaging[a] | |
| | Executing[b] | |
| | Innovation participants[a] | |
| | Key stakeholders[a] | |
| **Inductive domain** | **Inductive code** | **Theme** |
| Integration | Counseling | Integrated interventions |
| | Education and awareness | |
| | Group sessions | |
| | Home visits | |
| | Medication | |
| | MH screening | |
| | Monitoring | |
| | Referrals | |
| | Sharing experiences | |
| Mental health perceptions | | Local attitudes and perceptions of mental health conditions |
| Pathways | Diagnosis | Pathways to seeking care |
| | MH experiences | |
| | Posttreatment | |
| | Pretreatment evaluation | |
| | Treatment | |

[a]Constructs from CFIR 2022.
[b]Constructs from CFIR 2009.
[c]Constructs from Means *et al.*

'This is also important. There may be initial warning signs or early signs related to mental health or illness, and if your focus is on a psychiatric counselor or psychologist, they can pick up on these signs immediately. Early signs of depression, early signs of psychosis, and the like can be identified in this way, even if there is not previous history. Every single counseling session is important in this regard'. – MHP04

**Table 2.** Summary of total participants enrolled and related demographic information

| Type of participants | Number of IDIs conducted | Median age | Gender | |
|---|---|---|---|---|
| | | | Male | Female |
| TB survivors (TBS) | 10 | 35 (20–59) | 3 | 7 |
| Family members (FM) | 05 | 40 (32–59) | 3 | 2 |
| TB providers (TBP) | 05 | 42 (35–55) | 4 | 1 |
| MH providers (MHP) | 05 | 35 (32–50) | 3 | 2 |

**Table 3.** Mental health screening and associated TB units for TB survivors and their family members

| Participant ID | Participant type | Gender | Tuberculosis unit (TU) | MH condition screening |
|---|---|---|---|---|
| TBS01 | TB survivors with comorbid MH conditions | Female | Sahkarnagar | Depression; PHQ–9 score 17 |
| TBS02 | | Female | Sahkarnagar | Depression; PHQ–9 score 12 |
| TBS03 | | Female | Sahkarnagar | Self-reported depression |
| TBS04[a] | | Male | Jijamata (Pimpri) | Self-reported MH condition |
| TBS05 | | Female | Jijamata (Pimpri) | Depression and anxiety; PHQ–9 score 16 and GAD–7 score 15 |
| TBS06[a] | | Male | Jijamata (Pimpri) | Self-reported MH condition |
| TBS07 | | Male | Jijamata (Pimpri) | Depression and anxiety; PHQ–9 score 25 and GAD–7 score 19 |
| TBS08 | | Female | Sahkarnagar | Depression and anxiety; PHQ–9 score 11 and GAD–7 score 12 |
| TBS09 | | Female | Paud | anxiety; PHQ–9 score 9 and GAD–7 score 15 |
| TBS10 | | Female | Sahkarnagar | Self-reported depression |
| FM01 | Family member | Female | Sahkarnagar | N/A |
| FM02 | | Male | Sahkarnagar | |
| FM03 | | Female | Jijamata (Pimpri) | |
| FM04 | | Male | Paud | |
| FM05 | | Male | Sahkarnagar | |

[a]Specific MH condition, whether depression or anxiety, is unknown.

All participants also acknowledged the importance of survivors sharing their experiences with current patients. This theme is explored in the subsequent section.

### Agent for intervention delivery

Codes for intervention delivery were categorized into two sub-themes: (1) TB survivors delivering the intervention, or a part of the intervention, in a peer-led approach, and (2) other relevant stakeholders delivering the intervention. Three deductive codes were applied using CFIR: Innovation Deliverers, Innovation Participants and Engaging.

Suggestions included engaging TB survivors to share their experiences with current PWTB or engaging stakeholders, such as trained MH professionals, National TB Elimination Program (NTEP) staff or community workers to deliver the integrated intervention.

#### Peer-led support

TB survivors can be engaged in the intervention by sharing their experiences with the current PWTB. Participants explained that PWTB might find it comforting to hear experiences of individuals who have successfully completed TB treatment.

*'for sure, they should come forward and share things with them like I had these things [TB disease], then I received these treatments, and I recovered, so you also do it, so you can recover'. – TBS09*

*'…What happens most of the time, we become anxious thinking what will happen to us after four-five months? But when we see people who went through same phase and they get better/recovered, then we feel assured that this [recovery] can happen with me too. So, if this way we involve these patients in this then it will definitely be beneficial'. – MHP05*

#### Engaging health workers

Participants also talked about other key stakeholders who can be engaged in delivering such interventions: teams of trained MH providers, hospital staff, NTEP staff and non-governmental organizations.

*'psychiatrist, psychologist, social worker, and psychiatric nurse are four pillars of mental health. So, in any hospital these four pillars should be there, because mental health is also teamwork…'– MHP03*

*'like it is provided in government hospitals, it is okay, but along with that I feel it should be provided in societies, slum areas, it is needed to talk about this, conducting workshops in temples, Anganwadis [community health centers], then only people will come to know about it'. – FM03*

#### Importance of training intervention deliverers

MH providers expressed the importance of receiving training on managing PWTB so that they are able to understand patient experiences, and conveyed similar thoughts about TB providers receiving some training on identifying MH conditions.

*'…according to my information, TB patients mainly get medicines from DOTS providers [NTEP staff]. If we could train them [DOTS providers] to screen and identify patients [with MH conditions] like how to administer questionnaire, if we can do this then we can screen patients at that level to identify who needs mental health services' – MHP05*

### Local attitudes and perceptions of MH conditions

Two codes in this study were attributed to identify insights on stigma associated with MH or TB and perceptions of MH conditions. The deductive code, local attitudes, from CFIR was applied to

statements about MH or TB stigma. An inductive code, MH perception, was applied to general perceptions or misconceptions about MH conditions.

### Stigma associated with MH and TB

Most participants spoke about stigma related to MH conditions and TB disease. Some participants also explained how stigma associated with these conditions can exacerbate social isolation and, therefore, contribute toward worsening MH conditions.

*'Otherwise, I did not feel like going among people because if they come to know [about TB] then they will go away from me. And that is the reason I was not going out. Until I had my medicines, I did not go out with the fear that what if people will know. Doctors [TB health workers] also used to meet me and I used to think what if my neighborhood come to know. They will discuss about me. If one person come[s] to know…whole of village will know. Due to fear of these people, I stopped going out. That is the reason I was scared and never used to go out of my house'. – TBS07*

Providers spoke about how stigma can act as a barrier to seeking care.

*'…not having awareness, or stigma attached to a visit a psychiatrist. People may say, why you are going to the doctor of mad people [psychiatrist], why do you need him? You take TB tablets, and you will feel better, no need to visit a psychiatrist. So, this stigma is one of the sociocultural factors'. – MHP01*

### Familial perceptions and support

Family members and TB survivors expressed familial perceptions of MH conditions.

*'…See how it is, if the doctor explains well to the patient, and if the patient listens to it carefully, and if the family provides support to the patient, then there is no need to consult mental health professionals!'– FM05*

*'I did not show it to anybody because my family members said disease is a disease…there is nothing like mental disease. My grandmother said there is nothing like tension…' – TBS05*

*'Going to a psychiatrist is like I [had] gone mad for them [in-laws], they used to demean me for it…' – TBS10*

TB survivors also spoke about the importance of family support and the effect of familial support on their treatment journeys.

*'need to talk, need to explain, about TB. Need to make the family understand, because what happens at that time because of family, patient feels demotivated'. – TBS10*

### Journey maps of patient experiences with seeking MH and TB treatment

Inductive codes were developed to capture personal experiences with TB and MH. The parent code 'pathways' was used, with sub-codes such as 'treatment', 'diagnosis' and 'post-treatment' to record pathways.

Personal experiences with TB diagnosis and treatment, and post-TB experiences from our study are depicted below in the form of patient journey maps. The journey maps portray patient journeys with TB disease and MH condition based on the information gathered through the interviews. The maps (Figures 1 and 2) are not representative of all PWTB in Pune or India.

Figure 1 illustrates the journey of a female TB survivor who was initially treated for cough at a private facility and later diagnosed with TB at a public-sector facility upon a relative's advice. The diagnosis caused significant emotional distress, including fears for her life and her newborn's future, suicidal thoughts and family rejection. Despite completing treatment with support from TB providers, she remains anxious about the possibility of recurrence.

Figure 2 illustrates the journey of a male TB survivor who experienced suicidal thoughts and used alcohol to cope with his MH during treatment, leading to poor adherence and a second TB diagnosis. Fear of stigma kept him from disclosing his diagnosis, and he believes only curing TB – not MH services – can improve his condition.

### Discussion and recommendations

The goal of this exploratory study was to gather suggestions from TB survivors with lived experience of MH conditions, their family members and TB and MH health providers about design characteristics of potential integrated care models for MH and TB in India.

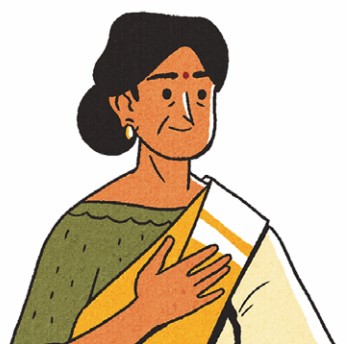

## Sakshi
(pseudonym)

**Tuberculosis unit**: Sahkarnagar, Pune, India

**Mental health condition screening:** Sakshi scored 17 on PHQ-9 and 12 on GAD-7

**Sakshi's journey with TB disease and mental health condition**

Visited a private health facility when she experienced persistent coughs. She found her way to a public sector facility at the suggestion of a relative. Here, she was diagnosed with TB.

Sakshi's diagnosis left her feeling stressed as she feared for her life and the life of her newborn child. Sakshi experienced suicidal thoughts and tried her best to stay positive.

Her family asked her to move back to her maidan home. They did not allow her to consume meat or eggs.

TB health providers offered her guidance and assured her that TB is curable. They asked her to communicate if she felt anxious or tense.

After completing her treatment, Sakshi continues to feel anxious about TB recurrence. Her body remains weak.

**Figure 1.** Patient journey map of a female TB survivor.

## Subhash

(pseudonym)

**Tuberculosis unit**: Jijamata, Pune, India

**Mental health condition screening:** Subhash
scored 25 on PHQ-9 and 19 on GAD-7

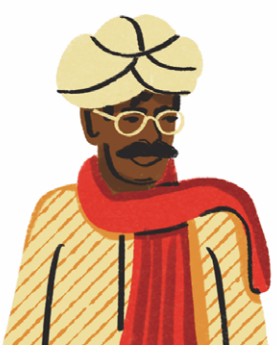

**Subhash's journey with TB disease and mental health condition**

When Subhash was diagnosed with TB after persistent cough, he feared his diagnosis. He didn't think he would survive. He experienced suicidal thoughts. He felt better after talking to his providers.

To manage his mental health condition, Subhash consumed alcohol when he was on TB treatment. He missed adhering to treatment and did not take his medicines regularly.

After 5-6 months, he was diagnosed with TB again.

He feared that his children would contract the disease since they lived in the same household. He did not tell anyone about this TB diagnosis due to stigmatization in the community.

He believes that his fear and anxiety cannot be managed by MH medications or counseling. He believes that only a cure from TB disease will alleviate his MH condition.

**Figure 2.** Patient journey map of a male TB survivor.

Interviewing participants after TB treatment completion allowed for reflection on the entirety of their TB care experience, including MH challenges and needs, throughout the TB care continuum (Venkatesan et al., 2025). This approach facilitated comprehensive and retrospective accounts, which helped surface critical insights into lived experiences. Further, intentionally recruiting TB survivors with self-reported MH conditions facilitated the inclusion of perspectives that may have otherwise been overlooked – particularly in settings where access to formal MH services is limited (Saraceno et al., 2007). Relying on MH diagnostic confirmation as an inclusion criterion could have excluded TB survivors without access to diagnostic services. This approach aligns with the study's broader aim to understand lived experiences and reflects the real-world conditions under which integrated TB–MH services must be implemented.

While it may have been ideal to interview TB patients with MH conditions immediately after treatment completion, the design of this study – embedded within a larger active trial – shaped our approach. Nevertheless, it allowed us to identify that participants continued to experience MH conditions even 18 months after treatment. The findings also indicated that some participants faced MH challenges not only during TB treatment but also well after its completion, underscoring the need to better understand these conditions in the context of post-TB sequelae.

Our results indicate that a package of interventions that can be implemented in the existing TB care program may be the most feasible strategy. Some providers advocated for a tailored approach toward intervention development, emphasizing that each PWTB and MH may have different needs.

According to the National Family Health Survey from India, 15.9% of women and 19.9% of men wanted a family member's TB disease kept secret. While most believed that TB can be cured, more than 50% had misconceptions about TB transmission (International Institute for Population Sciences (IIPS) and ICF, 2017). The results from our study align with the findings from this survey. TB survivors and their family members from our study shared similar sentiments about keeping TB disease a secret. Participants in our study also shared their misconceptions about TB disease transmission, treatment and cure. Further, a survey of

National Tuberculosis Program directors from high TB burden countries highlighted that the main perceived barriers to service integration were limited capacity, not recognizing MH as a problem and TB-related social stigma (Sweetland et al., 2019).

The National Mental Health Survey of India (2015–2016) (National Institute of Mental Health and Neuro Sciences Bengaluru, 2016) highlighted the potential of engaging nonspecialist professionals in the delivery of MH care. The survey revealed a scarcity of MH professionals, including psychiatrists, psychologists and psychiatric social workers, across the country. Given the limited availability of trained professionals, a peer-led approach may prove a feasible intervention.

An aim of the study was to use an implementation research framework to create an intervention that can be applied in Indian settings. With this aim in mind, a limited number of participants spoke about the long-term sustainability of their suggested interventions. Intended outcomes of interventions or programs may suffer if long-term sustainability is not considered. With significant costs and resources needed to initiate an integrated MH program, without thoughtful considerations for sustainability, the intervention may never reach full fruition. Additionally, new programs or interventions may not receive community or user support immediately (Shediac-Rizkallah and Bone, 1998). Thus, future research should consider the long-term effectiveness of the suggested interventions and parallel implementation assessments of acceptability and feasibility to provide evidence for scale-up of successful interventions.

Further, it is crucial to consider potential financial models for both the initial and long-term implementation of integrated services, particularly within the framework of India's national TB program. NTEP emphasizes the importance of collaboration with nongovernmental organizations and private healthcare providers. The Central TB Division of India oversees the financial management of the national program, which is supported by central government funding, as well as contributions from external partners, including international donors (Wells et al., 2024; Central TB Division, Ministry of Health and Family Welfare, 2024). Integrating MH services into the existing TB program may have implications for current benefits offered to patients. Integration could prompt broader discussions around expanding or adjusting the scope of

patient support packages to include MH-related costs – such as counseling, medication or transportation for MH appointments.

This research study may also have invariably raised awareness about MH conditions among participants, such as TB survivors and their family members, through the process of data collection. Participants seemed to have a growing understanding of MH and MH-related interventions as each interview progressed. Toward the end of each interview, PWTB and their family members had a better understanding of concepts such as stigma and MH conditions. Studies (MacNeill et al., 2016; Stelson et al., 2021) that examined research participation effects and unintended consequences of research participation reported that data collection processes may prompt participants to introspect and occasionally modify behaviors.

Research has shown that stakeholder engagement and participation can facilitate an enhanced contextual understanding that supports implementation (Murphy et al., 2021), increased buy-in to the research process (Kwon et al., 2017) and improved acceptability of potential intervention components (Marrone et al., 2022). The strengths of this study lie in its ability to capture multiperspective suggestions and experiences from a diverse range of participants, including patients, caregivers and healthcare providers. By engaging multiple stakeholder groups, the study gathered unique insights into the components of the intervention.

Suggestions for integrated interventions in this study are centered on providing MH care to PWTB. Achieving this requires an understanding of how PWTB navigate pathways to care. Understanding patient pathways can offer insights into how individuals might interact with the integrated interventions, and factors that could influence uptake and engagement (Design for Health, n.d.). A review of people-centered and participatory approaches in low- and middle-income countries shows that journey maps were often used in early explorations of lived experiences (Kang et al., 2025). In MH research, some qualitative studies called for the development of patient journey maps to visualize care-seeking modalities, while others used similar tools to synthesize qualitative findings (Lee et al., 2024; Antonaccio et al., 2025).

Recommendations for integrated interventions and related programmatic considerations for implementation include:

- Screening PWTB for MH conditions at the time of TB diagnosis and a few months after they initiate TB treatment.
- Capacity building efforts to train and sensitize MH providers about TB disease.
- Capacity building and sensitization of TB providers toward MH conditions that their patients may experience, and how to treat these conditions.
- Inviting TB survivors to share their TB or MH experiences with current PWTB and their family members.
- Offering TB and MH services at the same location, possibly at the same time, so that PWTB do not have to travel long distances.
- Tailoring MH referral and treatment processes based on MH diagnosis.
- Evaluating the sustainability of suggested interventions while considering India's health system and the NTEP.

## Limitations

This study has several limitations. MH conditions of all participants were not screened using one standardized method. A range of screening methods, such as PHQ-9, GAD-7 and self-reported MH conditions, were used. Inclusion of participants with self-reported MH challenges may encompass both normative stress

responses and clinically significant conditions. This study did not include a mechanism to recruit individuals who had completed MH treatment, which may limit insights into post-MH treatment experiences. However, the objective of this study was not to diagnose conditions but to capture perspectives from individuals who experienced MH challenges in the context of TB treatment. Additionally, since this study was embedded in the TB Aftermath trial, we only captured experiences and recommendations from participants enrolled or affiliated with the TB Aftermath trial, which may differ from those of survivors not in a TB study. However, it should be noted that TB Aftermath enrolled TB survivors at the end of TB treatment; thus, their MH experiences during TB treatment were not influenced by the trial. Additionally, this study may be subject to recall bias, as participants were asked to reflect on past experiences. The study also has limited generalizability as it only captured perspectives of TB survivors and providers from one Indian city. By focusing data collection and analysis efforts on CFIR, the study may have overlooked information that is beyond the framework's scope. Additional research on feasibility, acceptability and sustainability of the integrated care package may be required, as this study offers exploratory perspectives on MH, TB and integrated interventions.

## Conclusion

Working with people with lived experiences provides unique perspectives into MH and TB integration. TB survivors and their family members described their experiences in living with TB and MH conditions. Health providers shared their experiences in treating patients with either or both conditions. Stigma and community perspectives associated with MH and TB were highlighted as barriers to seeking care. Participant suggestions for integrated interventions included capacity building efforts for health providers, peer-led group sessions for TB survivors and their family members and regular counseling sessions for PWTB along the cascade of care. Understanding patient journeys and current health systems may provide valuable insights for intervention implementation and sustainability.

**Open peer review.** To view the open peer review materials for this article, please visit http://doi.org/10.1017/gmh.2025.10123.

**Supplementary material.** The supplementary material for this article can be found at http://doi.org/10.1017/gmh.2025.10123.

**Data availability statement.** All data are available upon request. The codebook used for analysis, along with coding frequencies, is provided in Supplementary Annexure 1.

**Acknowledgments.** The authors would like to acknowledge the contributions of all study participants including TB survivors, their family members, TB providers and MH providers for sharing their experiences. The authors would also like to acknowledge Sayali Sarpotdar, who served as a notetaker and transcriber for select interviews.
The journey maps and graphical abstract were designed using Canva.

**Author contribution.** Manvi Poddar: Data curation, formal analysis, visualization, writing – original draft. Madhuri Thorat Nalavade: Investigation, data curation, writing – review and editing. Nishi Suryavanshi: Conceptualization, project administration, writing – review and editing. Jonathan E. Golub: Conceptualization, supervision, writing – review and editing. Judith Bass: Conceptualization, supervision, writing – review and editing. Christopher Kemp: Conceptualization, methodology, supervision, writing – review and editing. TB Aftermath Study team: Project administration and resources.

**Financial support.** This study was funded by the Gupta-Klinsky India Institute at Johns Hopkins University (Breakthrough Research Grant) and the TB

Aftermath Study, which is funded by National Institute of Allergy and Infectious Diseases of the National Institutes of Health (grant number R01AI143748).

**Competing interests.** The authors declare none.

**Ethics statement.** This study was conducted in accordance with the ethical principles outlined in the Belmont Report. Ethical approval for this study was obtained from the Institutional Review Boards at Johns Hopkins University School of Medicine, Baltimore, MD, USA (FWA00005752, IRB00247239) and Dr. DY Patil Vidyapeeth, Pune, India (FWA00027671, IRB00011883). Written informed consent was obtained from all study participants before data collection. All participant identities were de-identified. Personal identifiers were removed from the data, and unique participant IDs were assigned to maintain anonymity throughout data analysis and reporting.

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
