## [Reviewer Report]

This study describes a qualitative exploration of the recalled experience of TB survivors with current mental health symptoms or retrospectively self-reported mental health symptoms during TB treatment, 18 months after completing treatment. It also examines the perspectives of family members, and TB and mental health providers. The conclusions are consistent with a large body of global qualitative data suggesting that TB and mental health are both highly stigmatized conditions, there is very limited community awareness of mental health issues and their treatments, routine screening for mental health conditions among people with TB is important, sensitizing TB and mental health providers about the other condition would be advantageous, and that integrated care is ideal. Although the authors did an adequate job applying the CFIR framework to inform, organize and present the findings, the title of the paper vastly overstates the findings as presented, and the “patient journeys” do little to elucidate pathways to care. Some TB survivors suggest that having seen a model of someone cured while in treatment might have been helpful to alleviate anxiety, this is far from suggesting that the best solution to addressing mental health among TB patients would be a peer-led solution. Very little distinction is made between what could be considered a normal reaction of stress or fear after having been diagnosed with a life-threatening illness and the type of disabling mental health conditions that require formal treatment. There is also insufficient justification for choosing TB survivors, a year and a half after completing treatment, as key informants to describe the mental health needs of people during TB treatment. To the contrary, given that mental health is the leading cause of disability for people post-TB, there was a missed opportunity to understand the long-term trajectories of mental health sequelae among TB survivors, long after treatment is over. Many of the TB survivors had severe depression and anxiety symptoms at the time of the interview (from 12 to 25), yet current episode did not appear to be explored, nor perceived attribution to the experience of having had TB. In sum, this paper does not make a substantive or novel contribution to the literature and, to the contrary, the overstatement of findings may be misleading.

---

## [Reviewer Report]

This article makes a valuable contribution to the understanding and implementation of integration of healthcare services. It is well written and provides a clear description of the research procedures. I raise two main points for consideration.

1) The authors do not specify the type of qualitative research they have conducted (for example, generic qualitative research, grounded theory, phenomenology, ethnography, etc). This appears to be a generic qualitative research. Although authors often omit that information in the published articles, it contributes to the transparency of the study to explicitly state its design. Similarly, the approach to the analysis could be more explicitly named as well. Is it template analysis (based on the CFIR), content analysis, thematic analysis, or another method?

2) The discussion section effectively highlights the significance of the major findings, which is important. However, it would benefit from a deeper engagement with the existing literature on the subject either in India or in other similar settings.

---

## [Editor Report]

Please review the paper based on comments made by the reviewers. While limitations have been described please discuss any limitations due to study design, timing of interviews in light of policy implications too.